# Antiviral Activity of Active Materials: Standard and Finger-Pad-Based Innovative Experimental Approaches

**DOI:** 10.3390/ma16072889

**Published:** 2023-04-05

**Authors:** Lea Szpiro, Clara Bourgeay, Alexandre Loic Hoareau, Thomas Julien, Camille Menard, Yana Marie, Manuel Rosa-Calatrava, Vincent Moules

**Affiliations:** 1VirHealth SAS, Innovation Centre Lyonbiopole, 321 Avenue Jean Jaurès, 69007 Lyon, France; 2CIRI, Centre International de Recherche en Infectiologie, (Team VirPath), Université de Lyon, Inserm, U1111, Université Claude Bernard Lyon 1, CNRS, UMR5308, ENS de Lyon, 69007 Lyon, France; 3Joint Technology Research Laboratory on Pathogenic Respiratory Viruses (PRV TEchLab), Innovation Centre Lyonbiopole, 321 Avenue Jean Jaurès, 69007 Lyon, France; 4VirNext, Faculté de Médecine RTH Laennec, Université Claude Bernard Lyon 1, Université de Lyon, 69008 Lyon, France; 5International Associated Laboratory RespiVir, Université Claude Bernard Lyon 1, 69008 Lyon, France; 6International Associated Laboratory RespiVir, University Laval, Québec, QC G1V 0A6, Canada

**Keywords:** active material, virus, antiviral activity, high-touch surface, finger-pad transfer

## Abstract

Environmental surfaces, including high-touch surfaces (HITS), bear a high risk of becoming fomites and can participate in viral dissemination through contact and transmission to other persons, due to the capacity of viruses to persist on such contaminated surface before being transferred to hands or other supports at sufficient concentration to initiate infection through direct contact. Interest in the development of self-decontaminating materials as additional safety measures towards preventing viral infectious disease transmission has been growing. Active materials are expected to reduce the viral charge on surfaces over time and consequently limit viral transmission capacity through direct contact. In this study, we compared antiviral activities obtained using three different experimental procedures by assessing the survival of an enveloped virus (influenza virus) and non-enveloped virus (feline calicivirus) over time on a reference surface and three active materials. Our data show that experimental test conditions can have a substantial impact of over 1 log_10_ on the antiviral activity of active material for the same contact period, depending on the nature of the virus. We then developed an innovative and reproducible approach based on finger-pad transfer to evaluate the antiviral activity of HITS against a murine norovirus inoculum under conditions closely reflecting real-life surface exposure.

## 1. Introduction

One particularly significant factor in the transmission of viral infectious diseases is the ability of viruses to persist on surfaces in both healthcare and everyday settings. Several reviews and models have suggested that indirect contact transmission involving contaminated surfaces (fomites) could be the predominant transmission route for certain respiratory and enteric viruses [1,2]. In fact, a recent study uncovered surface contamination as a more significant spreading route for many diseases than previously believed [3]. Surfaces become contaminated either through the deposition of virus-containing respiratory droplets emitted from an infected individual via coughing/sneezing, or via direct contact with contaminated hands or items with virus excreted from patient secretions/excretions (bronchoalveolar fluid, sputum, mucus and saliva). Similarly, non-enveloped gastroenteritis viruses such as norovirus and rotavirus are well known to persist for several weeks on many different types of surfaces contaminated through contact with soiled hands due to unhygienic practice, causing large outbreaks in both healthcare facilities and communities [4,5]. High-touch surfaces (HITS), such as door handles, bear a high risk of becoming fomites and can participate in viral dissemination through contact and transmission to other persons. The relevance of this pathway is supported by experimental transmission studies in animal models and by the results of investigations on HITS contamination, with such surfaces representing over 50% of those contaminated by enveloped and non-enveloped viruses in healthcare settings, homes and daycare centers [6,7]. In order to consider contaminated surfaces as playing a role in transmission, viral pathogens shed into the environment must have the capacity to persist on such surfaces before being transferred to hands or other support at sufficient concentrations to initiate infection through direct contact with the eyes, nose or mouth, for example. An analysis of COVID-19 superspreading events in Austria and mathematical modelling estimated that between 100 and 1000 (2 to 3 log_10_) infectious virions were sufficient to cause infection [8,9]. Such number has been confirmed using animal models of COVID-19 infection [10]. Previous reports have suggested that the minimal dose of norovirus required to cause human infection (severe diarrhea and dehydration) could be between 10 and 100 infectious viral particles [4,11,12]. Laboratory simulations have demonstrated that hand-to-hand and fomite-to-hand contact are viable modes of transmission for a large panel of viral strains [13,14] provided that viral particles remain viable on hands and fomites in sufficient quantities. Data demonstrating the transfer of human parainfluenza virus, influenza virus, rhinovirus and norovirus from contaminated surfaces to clean fingers support such a role for fomites in the contamination of hands [4,15]. The potential of a surface/fomite to participate in the spread of a given infectious virus is thus directly correlated to infectious virus persistence capacities until it reaches respiratory or digestive tracts via finger contamination. The surface stability of a virus generally depends on its intrinsic nature (enveloped versus non-enveloped viral strain), surface factors (chemical composition, roughness, porosity, surface absorption and hydrophobicity), environmental factors (relative humidity (RH), temperature and exposure to light), and the presence of body fluid secretions (organic matrix) and other microorganisms [16,17]. Studies on the stability of viruses on surfaces in different laboratory-simulated environments have demonstrated the involvement of many experimental factors including the titer of the virus, contact area, volume of the inoculum, pH and presence of salts. In general terms, infectious enveloped viruses such as SARS-CoV-2 or influenza virus were shown to persist for up to 7 days on various smooth surfaces, whereas non-enveloped viruses such as calicivirus and hepatitis A virus could persist for up to several weeks.

In recent years, growing interest in the use of self-decontaminating surface materials as additional safety measures towards preventing disease transmission has emerged [18]. These materials are based on surface-bound active antimicrobials and biocidal coatings [19] or passive pathogen-repellent surfaces [20], and have been developed using micro/nanomaterials, chemical modifications and micro- or nano-structuring [21,22]. They are expected to reduce the viral charge on surfaces over time and consequently limit viral transmission capacity through contact. The underlying two-way mechanism involves the inactivation of virus particles via either their interaction directly with the surface or with ions released upon contact of the surface with a particular environment. The most studied antibacterial/antiviral nanomaterial, nano/micro-silver (Ag), owes its antimicrobial properties to the release of Ag+ ions upon contact with aqueous systems [23]. Silver deactivates viruses by targeting the viral envelope and viral surface proteins, thereby blocking the penetration of that virus into cells, and by interacting with the cellular pathways, viral genome and viral replication factors [24,25]. Silver is reported to inhibit several viruses, including SARS-CoV-2, herpes simplex virus 2, hepatitis B virus, Tacaribe virus, vaccinia virus, and H1N1 influenza A virus [26,27,28,29,30,31]. Similarly, copper nano/micro-particles have been studied as antibacterial materials, and copper oxide has demonstrated antimicrobial activity due to the generation of reactive oxygen species and potentially also surface-related catalytic activity [32]. While copper ion release appears essential to the maintenance of antimicrobial efficacy, the mechanism of action remains unclear [33]. The antiviral effect of copper has previously been reported for HCoV-229E, SARS-CoV-2 and orthopoxvirus [34,35,36]. Silver and copper ions present the broadest spectrum of antiviral activity among the different metal ions studied [37].

The antiviral activity (log_10_ TCID50/PFU per cm^2^) of active materials corresponds to the log_10_ difference between the infectious titer of a virus found on an untreated product versus an antiviral-treated product after inoculation or contact with viruses during a defined period. Among the limited number of experimental procedures reported, the international standard ISO21702, the counterpart standard of the ISO22196 for antibacterial activity, makes it possible to determine antiviral activity of an active surface onto which a viral inoculum in liquid state has been deposited for a defined contact period at 25 °C and at 90% RH [38]. Recently, the NFS90700 standard has been proposed as a method to simulate near-ambient test conditions and hand-borne contamination in order to assess antibacterial activity of active surfaces [39]. This experimental strategy involves depositing a weak inoculum onto a small area of the active surface, whereby its quick drying in less than 3 min and incubation at 20 °C and 40–60% RH means the inoculum remains completely dry throughout the contact period.

In our study, we took into account the impact of experimental and environmental parameters on both virus persistence on test surfaces and antiviral activity of active materials. For this, we compared antiviral activities obtained using three different experimental procedures by assessing the survival of an enveloped virus (influenza A virus) and nonenveloped virus (feline calicivirus) over time on a reference surface and three active materials. We then developed an innovative approach based on finger-pad transfer to evaluate the antiviral activity of high-touch active materials against a murine norovirus inoculum, a surrogate for human gastroenteritis viruses, under conditions closely reflecting real-life surface exposure.

## 2. Materials and Methods

### 2.1. Viruses and Cell Lines

Murine norovirus (MNV-1, ATCC VR-1937), feline calicivirus (FCV, ATCC VR-782) and human influenza virus (H3N2, ATCC VR-1679) were produced on specific cell lines: RAW 264.7 (ATCC TIB-71) for MNV-1, KE-R (CCLV-RIE 0138) for FCV and MDCK (ATCC CCL-34) for H3N2 virus. Cells were cultivated in DMEM 1 g/L, DMEM 4.5 g/L or in EMEM supplemented with L-glutamine, antibiotics, and a controlled quantity of FCS (10% or 5%). Virus production and quantification were performed in an infection medium (EMEM or DMEM with 2% or without FCS). For influenza virus, Trypsin acetylated from bovine pancreas at 2 µg/mL was added to the culture medium.

### 2.2. Active and Reference Materials

Stainless steel discs (1.4301, 5 cm × 5 cm with Grade 2 B finish on both sides) were used as a reference surface for both antiviral activity and fingerprint experiment. Stainless steel discs were functionalized, which included a DECON 5% bath and a 70% isopropanol bath interspersed with water bath. Stainless steel discs were sterilized by autoclaving for 15 min at 121 °C. AS1 material corresponds to a PVC membrane with a topcoat layer (PMMA acrylic and PVC resins) containing micro-silver glass phosphate particles (<15%). AS2 material corresponds to the 1.4 mm layer obtained by extraction, containing 82–87% PVC resin, 0.8–1% silver, 1–2% Ca/Zn heat stabilizer and 12–15% of additives (Escort/V, Gerflor, Lyon, France). The AS3 material corresponds to a 120 µm layer of a polyester/metal mix (90–93% copper) sprayed over stainless steel coupons (MetalSkin^®^, MetalSkin Technologies, Balaruc les Bains, France).

### 2.3. Antiviral Activity of Active Materials

Three protocols for the determination of antiviral activity were compared in this study. The first one corresponds to the ISO21702 standard [38]. A volume of 400 µL of viral inoculum (5.6.105 TCID50/mL) was deposited onto a 16 cm^2^ surface. The inoculum was covered by a PE film. After incubation for a defined contact time at 25 °C and 90% RH, the residual virus was harvested with 10 mL of maintenance medium. The residual virus on the surface was quantified using tissue-culture infectious dose 50% (TCID_50_) and the Spearman–Kärber technique. The adapted protocol (ISO21702AD) corresponds to the ISO21702 protocol, except that the inoculum was not covered by a film and the RH during incubation period was 41%. The NFS90700AD protocol consisted of depositing 20 µL (1.107 TCID_50_/mL) onto a 1 cm^2^ surface [39]. The inoculated surface was put under hood flow for quick drying in less than 5 min by streaking the inoculum with a pipet tip. After incubation for a defined contact time at 20 °C and 50% RH, the residual virus was harvested with 2 mL of maintenance medium. The residual virus on the surface was quantified using tissue-culture infectious dose 50% (TCID_50_) and the Spearman–Kärber technique.

### 2.4. Finger-Pad Experiment

A volume of 20 µL of viral suspension (1.107 TCID50/mL) was deposited onto 1 cm^2^ area of a middle fingertip previously cut from a nitrile glove and put under hood flow for quick drying by streaking the inoculum. Once dry, the fingertips were positioned onto the middle finger of a gloved hand of an operator and pressed on to the test surfaces for 10 s with one rotating movement of the finger (90°) and a mechanical pressure of 1 kg controlled using a scale. The surfaces where the fingertips made contact were washed in maintenance medium and the virus remaining on contaminated nitrile-gloved middle fingertips was recovered by sonication procedure (10 min, 40 Hz). The samples were then prepared for quantification utilizing tissue-culture infectious dose 50% (TCID_50_). The residual virus was quantified by Spearman–Kärber methods. For the deposit of complex inoculum, 20 µL of mixture (V/V) containing murine norovirus (1.33.107 TCID_50_/mL) and Enterococcus faecium bacteria (4.56.107 CFU/mL) was deposited onto a 1 cm^2^ area. The surfaces where the fingertips made contact were washed in 2 mL of maintenance medium. Half of the volume was used for bacterial quantification on Tryptone soy agar (TSA Merck, R n°105458). The CFU were quantified after 24–48 h at 37 °C. The other half of the volume was centrifuged to eliminate the bacteria and used for viral quantification using tissue-culture infectious dose 50% (TCID_50_).

## 3. Results

### 3.1. Impact of Experimental Parameters on the Survival of Enveloped and Non-Enveloped Viruses on the Stainless Steel Reference Surface

We aimed to investigate the impact of different experimental parameters used in three antiviral activity protocols (ISO21702, ISO21702AD and NFS90700AD), including virus deposition- and environmental-specific parameters, on viral persistence on a particular surface. To this end, we first tested the impact of these parameters on the survival of both enveloped H3N2 influenza virus and the non-enveloped feline calicivirus on stainless steel surfaces, used as the reference support to determine the antiviral activity of active material over a defined period of time. We prepared one group of inoculated supports according to the ISO21702 protocol, corresponding to the deposition of 400 µL of viral solution (2.10 × 10^5^ TCID_50_) onto a 16 cm^2^ area, then left the inoculated surfaces in an incubator at 25 °C, half with a RH of 90% (ISO21702 protocol, inoculum covered with a PE film) and the other half at a RH of 50% (ISO21702AD protocol). At 90% RH, the inoculum remained in a liquid state throughout the incubation period, while the viral inoculum completely dried after an incubation period of 5 to 6 h post application at 50% RH (data not shown). We also prepared a second group of inoculated supports according to the NFS90700 modified protocol (NFS90700AD), corresponding to the deposition of 20 µL of viral solution (2.10 × 10^5^ TCID_50_) onto a 1 cm^2^ surface area with drying through mechanical contact with pipette tips, allowing the complete drying of the inoculum in 5 min post application. Inoculated supports were then left in an incubator at 20 °C with an RH of 50% (NFS90700AD protocol). For each test surface and experimental condition, inoculated supports were retrieved at desired time points (0, 5, 15 and 60 minutes and 2, 8, 24 and 48 hours post application) and immediately soaked with viral culture media before determining the infectious viral titer (TCID_50_) (Spearman and Kärber Methods). To compare viral persistence on stainless steel using different experimental conditions, biphasic linear regression plots (log_10_ titer vs. time) of the survival data were used to calculate the half-life (t^1/2^, time required to reduce the viral titer by one-half), the D value (duration of time required to reduce the initial burden by 1 log_10_) and the residual number of infectious viruses on the support at 5, 15, 30 and 60 min and 2, 8, 24 and 48 hours post application. We reported all experimental measurements as means of 18 replicates with SD. A Student statistical test with a confidence interval of 95% was carried out to evaluate significant differences among the data (*p* < 0.05 statistically significant).

On the stainless steel support, H3N2 influenza virus showed high stability with no significant difference in the residual amount of virus on the support at 60 min post application for the three protocols (Figure 1a). However, at 2 h and 8 h post application, the amount of virus present on the surface was significantly lower according to the NFS90700AD protocol compared to the other two (*p* < 0.0001 and *p* = 0.01, Figure 1b). An average loss of 2 log_10_ TCID_50_ was measured at 48 h post application, with a significant statistical difference observed between ISO21702AD and NFS90700AD parameters only (*p* = 0.01). For H3N2, among all experimental parameters from the ISO21702 and ISO21702AD protocols, only the amount of virus at 24 h post application was found to be significantly different (*p* = 0.039). However, the amount of virus at 8 h (*p* = 0.005) and 24 h (*p* = 0.005) post application, half-life (*p* = 0.001) and D (*p* = 0.001) values were significantly different between ISO21702 and NFS90700AD protocols (Figure 1b). Values obtained for all parameters, including the amount of virus at 48 h post application (*p* = 0.01), were significantly different between ISO21702AD and NFS90700AD protocols. As observed for H3N2 virus, feline calicivirus showed high stability with no significant differences among the three experimental procedures at 60 min post application (Figure 1c). The calicivirus showed similar surface stability over time on the stainless steel support using ISO21702 and ISO21702AD protocols, with most parameters showing no significant differences; only the amount of virus at 8 (*p* = 0.005) and 24 h (*p* = 0.036) post application was significantly different between the two protocols, associated with an average loss of 1.6 log_10_ TCID_50_ at 48 h post application (Figure 1d). NFS90700AD experimental parameters showed the strongest impact on calicivirus persistence over time, with significant differences in the values obtained as compared to those using either ISO21702 or ISO21702AD experimental parameters, namely, the amount of virus at 2 h (*p* < 0.001/*p* < 0.0001), 8 h (*p* < 0.0001/*p* < 0.0001), 24 h (*p* = 0.005/*p* < 0.0001) and 48 h (*p* < 0.001/*p* = 0.013) post application, half-life (*p* = 0.001/*p* = 0.018) and D (*p* = 0.001/*p* = 0.006). A further loss of around 0.8 log_10_ TCID_50_ was observed at 48 h post application using the NFS90700AD protocol as compared to the other two protocols (Figure 1d).

### 3.2. Residual Infectious Virus on Surfaces over Time and Antiviral Activities of Active Materials According to Experimental Protocols

We next evaluated the amount of residual H3N2 or feline calicivirus infectious viruses on each of three inoculated active materials (AS01, AS02 and AS03) over time (Figure 2 and Figure 3). As described above, one group of inoculated active supports was prepared according to the ISO21702 protocol, then incubated at 25 °C and either 90% (ISO21702) or 50% (ISO21702AD) RH. The second group corresponded to active supports inoculated according to the NFS90700AD protocol that were incubated at 20 °C and 50% RH. For each active surface and experimental condition (nine replicates per active surface/method), inoculated supports retrieved at desired time points (0, 2, 8, 24 and 48 h post application for AS01 and AS02 surfaces; 0, 5, 15, 30 and 60 min post application for AS03 surface) were immediately soaked with viral culture media and infectious viral titer (TCID_50_) was determined. The average antiviral activity Rm (log_10_ TCID_50_/cm^2^), including a 95% confidence interval (Kr) of the three active materials from three independent experiments, was then determined. Antiviral activity R corresponded to the difference in the logarithm of the infectivity titer of the virus found on an untreated product and an antiviral-treated product after a period of incubation and contact with the virus. Stainless steel supports were taken as the untreated (reference) surface.

The amount of H3N2 infectious virus on the AS01 surface reached the lower limit of quantification at 48 h post application, whereas no H3N2 infectious particles were quantified at 24 h post application on the AS02 surface using all three protocols (Figure 2a,c). ISO21702 and ISO21702AD protocols gave similar values for the amounts of virus observed on AS01 at both 2 and 8 h post application. As observed on stainless steel supports at 24 h post application, the H2N2 virus was statistically the most stable on AS01 under ISO21702AD experimental conditions (Figure 2a). The NFS902700AD protocol had an impact on the survival of the H3N2 virus on AS01, as shown by the significantly reduced amount of infectious virus over time as compared to those quantified using ISO21702 and ISO21702AD protocols. The same impact was observed at 15 min and 8 h post application for AS03 and AS02, respectively (Figure 2c,e). A difference of 1 log_10_ TCID_50_ was observed on AS03 at 15 min post application, between NFS90700AD and ISO21702 or ISO21702AD protocols (Figure 2e). The antiviral activities Rm (log_10_ TCID_50_/cm^2^) of the three active surfaces, corresponding to the difference in the logarithm of the infectivity titer of H3N2 virus found on stainless steel supports versus antiviral-treated, were determined over time. While the antiviral activity of AS01 at 2 h post application showed no significant difference among the three protocols, antiviral activities were significantly higher at 8 and 24 h post application using the NFS90700AD compared to the ISO21702/ISO21702AD protocols (Figure 2b). While all three protocols revealed similar antiviral activities for AS03 and AS02 at, respectively, 5 min and 2 h post application, a stronger antiviral activity for AS03 and for AS02 was revealed at, respectively, 15 min and 8 h post application using the NFS90700AD protocol (Figure 2d,f).

The residual amount of calicivirus on AS01 and AS02 reached the lower limit of quantification at 48 h post application according to all three protocols (Figure 3a,c). Similar amounts of infectious virus were observed on AS01 using ISO21702 and NFS90700AD protocols both at 2 and at 24 h post application (Figure 3a), whereas the ISO21702AD protocol had a lower impact on the survival of calicivirus on AS01 at 24 h post application. Similarly, while no significant difference was observed in the amount of virus on the AS02 support at either 2 or 8 h post application using the ISO21702 and NFS90700AD protocols (Figure 3c), the ISO21702AD protocol had a lower impact on the survival of calicivirus at 8 and 24 h post application. The lowest residual amount of virus was found on AS02 at 24 h post application using the ISO21702 protocol. While the amount of virus on the AS03 support was similar between the three experimental procedures at 5 min post application, the NFS90700AD protocol showed a greater impact on the viral viability at 15 min post application (Figure 3e). The antiviral activity Rm of the three active surfaces was determined over time (Figure 3). While no significant difference was observed among the three protocols with regard to the antiviral activity of AS01 at 2 h post application, values for antiviral activity according to the ISO21702 protocol were significantly higher than those obtained both using the NFS90700AD protocol at 8 and 24 h post application and using the ISO21702AD protocol at 24 h (Figure 3b). The antiviral activity at 24 h post application was significantly higher according to the ISO21702AD protocol compared to that obtained with the NFS90700AD protocol. Similarly, the most significant antiviral activity against calicivirus on the AS02 material was found at 8 and 24 h post application using the ISO21702 protocol, while the smallest antiviral activity was obtained using the NFS90700AD protocol at 24 h post application, corresponding to a decrease of 1 log_10_ in antiviral activity between ISO21702 and NFS90700AD protocols (Figure 3d). As observed for the H3N2 virus (Figure 2f), the highest antiviral activity of AS03 material at 15 min post application was obtained using the NFS90700AD protocol.

### 3.3. Antiviral Activity of High-Touch Active Surfaces According to a Finger-Pad Transfer Experimental Method

To better understand if virus transfer can occur from the touching of a contaminated fomite, we next developed an experimental procedure adapted from the finger-pad transfer experiment proposed by Bidawid et al. [4] to enable the evaluation of the antiviral activity of high-touch surfaces (HITS) under conditions closely reflecting real-life surface exposure. The inoculation involved the direct transfer of viruses from a contaminated gloved finger to reference or active surfaces. The use of gloves minimized ethical issues and biosecurity risk, ensured a high reproducibility of viral transfer and enabled the assay to be performed with BSL2 and BSL3 viral strains. In short, 20 µL of a virus-containing soiling solution (10^5^ TCID_50_) was quickly dried onto a 1 cm^2^ surface of the middle finger cut from a nitrile glove. Once dry, the contaminated fingertip was positioned onto the middle finger of a gloved hand of an operator and pressed onto the test surfaces for 10 s with one rotating movement of the finger (90°), associated with a mechanical pression of 1 kg at 20 °C and 40–60% RH. The surface where the fingertip made contact was washed in maintenance medium, and the residual amount of virus on the contaminated nitrile-glove fingertip was recovered by a sonication procedure. The samples were then prepared for the quantification of infectious viral titer (TCID_50_).

We first assessed the possibility of transferring murine norovirus (MNV-1), a surrogate for human gastroenteritis viruses, from gloved fingertips to a reference stainless steel surface (Figure 4). A soiling solution corresponding to a final concentration of 3 g/L of bovine serum albumin (BSA) was added to the viral solution as a moderate stool mimetic mixture. The quantity of infectious virus transferred onto the reference stainless-= steel surface and the residual amount of virus on the gloved finger after contact were determined after one, two or three successive contacts without additional contamination of the gloved finger (Figure 4a).

The average amounts of infectious virus isolated from the inoculated reference surface after the first and the second gloved finger contact were 3.82 ± 0.13 log_10_ TCID_50_ and 1.76 ± 0.18 log_10_ TCID_50_, respectively; the amount of virus isolated from the reference surface after the third contact was below the lower limit of quantification, but still detectable on the glove itself (Figure 4a). Based on these data, the transfer efficiency percentage of MNV-1 in our experimental conditions, according to Julian et al. [40], was 4.4% and 0.05% after the first and the second gloved finger contact, respectively. 

The survival of norovirus over time on the reference stainless steel support inoculated through finger-pad transfer or using the NFS90700AD protocol (10^4^ TCID_50_, 3 g/L of BSA) was also evaluated (Figure 4b). While the quantity of infectious norovirus on reference supports inoculated through the fast drying of a viral solution according to the NFS90700AD protocol remained stable at 60 min post application, we observed a loss of 1.3, 1.2, 1.6 and 1.7 log_10_ TCID_50_ at 5, 15, 30 and 60 min post application, respectively, from reference supports inoculated by finger-pad transfer of the norovirus/BSA mixture. After 30 min post application, the amount of norovirus on the finger pad-inoculated support remained stable for up to 60 min (Figure 4b).

We then determined the quantity of infectious norovirus remaining over time after a finger-pad transfer onto AS03 supports corresponding to a polymer composite containing over 90% solid copper alloy (Figure 5). We found similar quantities of infectious norovirus on the reference stainless steel and AS03 supports after finger-pad transfer at T0 (Figure 5a). However, the residual quantity of infectious norovirus decreased strongly on AS03 to reach the LLOQ by 15 min post application (Figure 5b). The amounts of infectious virus on the reference stainless steel and AS03 supports at 5 and 15 min post application differed by 1.00 ± 0.16 and 1.90 ± 0.13 log_10_ TCID_50_, respectively. The antiviral efficacy of the AS03 material was therefore 90.0% and 98.7% at 5 and 15 min, respectively. To understand transfer capacities and the antiviral efficiency of the active material in more realistic conditions, we evaluated the transfer efficiency of a complex mixture containing both murine norovirus (10^5^ TCID_50_) and *Enterococcus faecium* (10^5^ PFU) in the presence of 3 g/L of BSA, from gloved fingertips to both reference stainless steel and AS03 supports (Figure 6 and Figure 7). *Enterococcus faecium* is a common intestinal commensal bacterium that is responsible for a range of hospital- and community-acquired infections worldwide [41].

The average amount of infectious norovirus isolated from the reference surface inoculated through contact with a gloved finger contaminated with the complex mixture was 3.6.10^3^ TCID_50_ (3.55 ± 0.15 log_10_) (Figure 6a). This corresponds to a 4.5% transfer efficiency of MNV-1 in this experimental condition, similar to the one calculated without bacteria (4.4%, Figure 4a).

We next evaluated the persistence over time of the norovirus in the complex mixture transferred to the reference stainless steel support by finger contact (Figure 6b). At first, a difference of 0.5 log_10_ TCID_50_ was observed at 5 min following the transfer onto the stainless steel supports in the presence of bacteria as compared to the virus alone (Figure 6b, complex, *p* = 0.01). The virus survival profiles obtained from the two experimental conditions then become very similar over time, up to 60 min post application when the residual amount of virus on the reference surface was significantly greater in the presence of bacteria (*p* = 0.001). The average amount of *Enterococcus faecium* bacteria isolated from the inoculated surface after complex-mixture-contaminated gloved-finger contact was 4.2×10^3^ CFU (3.62 ± 0.21 log_10_) (Figure 6c), corresponding to a 13.4% transfer efficiency of bacteria in our experimental conditions. The amount of bacteria in the complex mixture transferred to the reference stainless steel support by finger contact decreased over time to reach 2.2 log_10_ CFU at 60 min post transfer (Figure 6d).

We next determined the residual quantity of both infectious norovirus and *Enterococcus faecium* over time after a finger-pad transfer onto AS03 active material (Figure 7). Similar quantities of infectious microorganisms were observed on reference stainless steel and AS03 supports at T0 following finger-pad transfer. The residual quantity of infectious norovirus decreased strongly over time on AS03 supports, nearly reaching the LLOQ by 15 min post transfer (Figure 7). The amount of virus observed on reference stainless steel and AS03 supports at 5- and 15 min post transfer differed by 0.98 ± 0.09 and 1.64 ± 0.15 log_10_ TCID50, respectively. The AS03 material thus demonstrated an antiviral efficacy of 89.5% and 97.7% at 5 and 15 min, respectively. The residual amount of norovirus on the AS03 support was significantly different at 5 (*p* < 0.0001) and 15 (*p* = 0.002) minutes post transfer between experimental conditions with or without bacteria (Figure 5b and Figure 7). However, the reduction R (log_10_ TCID50/cm^2^) at 5 min post transfer was not significantly different with or without bacteria. Similarly, the residual quantity of *Enterococcus faecium* bacteria after a finger-pad contact on the AS03 material decreased over time, almost reaching the LLOQ at 15 min post transfer (Figure 7). The number of CFU observed on the reference stainless-steel and AS03 supports at 5 and 15 min post transfer differed by 0.95 ± 0.08 and 1.65 ± 0.13 log_10_ CFU, respectively. The AS03 material thus demonstrated an antibacterial efficacy of 88.8% and 97.8% at 5 and 15 min, respectively. At 30 min post transfer, no detectable infectious norovirus or viable *Enterococcus faecium* bacteria were quantified on AS03 materials, whereas 2.16 ± 0.20 log_10_ TCID_50_ and 2.53 ± 0.04 log_10_ CFU were quantified on the reference stainless steel surface (Figure 7).

## 4. Discussion

The role played by contaminated surfaces in the transfer and spread of viruses is now well established [4,6]. Their potentially significant implication in hospital-acquired viral infections has gained importance since the COVID-19 pandemic. Surrounding surfaces and high touch surfaces (HITS) can be contaminated via deposits of airborne droplets containing respiratory virus emitted when infected individuals cough or sneeze, or via contact with hands soiled with respiratory secretions/excretions. Similarly, soiled hands of an individual infected with viral gastroenteritis, if not washed efficiently, participate towards the spread of this infectious disease within a community. The potential of a surface/fomite to allow the spread of a given infectious virus is directly correlated to the capacity of that virus to persist in sufficient quantities to reach the respiratory or digestive tracts via finger transfer. Virus persistence on inert surfaces is mainly influenced by the presence or absence of a viral envelope, the type of surface and environmental factors such as temperature and relative humidity (RH) [16,17,42,43,44]. Differences in surface stability can be observed in the scientific literature between two similar laboratory simulated experiments, which may be the consequence of differences in experimental parameters such as the titer of virus stock, the contact area, the volume of the inoculum and methods used to deposit/recover/titrate viral particles. Self-decontaminating surface materials formed by incorporating/coating with, for example, antibacterial/antiviral micro/nanomaterial may reduce the bacterial/viral charge on surface over time more rapidly than on a neutral surface, and consequently may limit bacterial/viral transfer by contact with objects or hands. The antiviral activities of such active materials over time can be calculated by comparing residual quantities of viruses on reference supports versus active materials during a defined period. We designed our study to compare three experimental methods of evaluating antiviral activity, differing in terms of deposition- and incubation-specific parameters, on three active materials over time using a stainless steel support as the reference. While the incubation temperature was similar for all three procedures (20–25 °C), parameter differences included the state of the inoculum on the support (dry versus wet), the surface area covered by the inoculum (1 vs. 16 cm^2^), the amount of virus deposited per cm^2^ (5.30 vs. 4.14 log_10_ TCID_50_), and the relative humidity during the incubation period (50 vs. 90%). While the three experimental procedures had only a moderate impact on virus survival on the stainless steel support over time, observed differences were found to mainly depend on the viral strain tested. As expected, differences in residual virus quantity on the stainless steel support between the enveloped influenza virus and the non-enveloped calicivirus were apparent at 48 h post application. This is in accordance with previous findings of infectious enveloped viruses persisting for up to 5–7 days, and non-enveloped viruses for up to several weeks, on various smooth surfaces [2,17,43]. The NFS90700AD protocol seemed to have a significant impact from 2 h post application on the survival of both enveloped and non-enveloped viral strains, the impact being most obvious over time on the feline calicivirus. The difference may be linked to the rapid mechanically assisted drying of the inoculum on 1 cm^2^ compared to gentle drying on 16 cm^2^ inoculum according to the ISO21702AD protocol. To date, no study has been performed describing the impact of inoculum size (surface area and quantity) on viral survival on smooth surfaces. Interestingly, virus survival profiles over time are similar between ISO21702 and ISO21702AD experimental conditions, except for the residual quantity of the virus at 8 h post application for calicivirus and at 24 h post application for both H3N2 and calicivirus. Multiple mechanisms relating to the level of RH (50 vs. 90%), including the desiccation and interaction of viral capsids at the air-water interface (AWI) of a solution, may contribute to viral inactivation on surfaces [45,46]. The antiviral activity according to the ISO21702 procedure is measured using an inoculum maintained in a liquid state throughout the incubation period. During the incubation of the micromaterial-containing active surface with the virus-containing inoculum, virus particles may be inactivated either by their direct interaction with the surface or by their interaction with the ions released upon contact with aqueous systems [23]. Considering the kinetics of virus particles versus the released ions in the liquid solution, virus particles display much slower motion than ions, indicating that any ion-related antiviral effect would occur faster [18,23]. During the NFS90700AD procedure, virus particles may be inactivated by their direct interaction with active micromaterials or with the released ions on the surface, although the diffusion of ions in the dry inoculum would be expected to be quite limited. The desiccation effect caused by the inoculum drying step and the incubation RH (50%) may contribute to the inactivation of a virus. The ISO21702AD protocol proposed in our study combines a first phase, in which the inoculum remains in a wet state (0 to 5 h post application, data not shown), and a second one during which the inoculum is completely dry. The strongest antiviral activities over time measured in our study on three active materials corresponded to the use of the NFS90700AD protocol for the H3N2 influenza virus. We expected silver micromaterial-containing surfaces to reach greater antiviral activity using the ISO21702 protocol due to the release of ions in the liquid phase, as described in several studies [18,23,47]. AS01 and AS02 supports are active materials in which the antiviral effector (silver) has been incorporated into the material mass and not just coated onto the surface. The quantification of the amount of released ions from AS01 and AS02 materials could potentially give explanations on moderate antiviral activity according to the ISO21702 protocol. The direct contact of the virus with active surface elements or ions closed to the surface, as well as the desiccation effect caused by the NFS90700AD protocol, had a stronger impact on the survival of enveloped viruses. Similarly, AS03 material that contained copper alloy presented the strongest antiviral activity against the H3N2 influenza virus at 15 min post application using the NFS90700AD protocol. Warnes et al. investigated the use of copper alloys for the inactivation of human coronavirus 229E and showed that the complete inactivation of 10^3^ plaque-forming units (PFU) applied to a 1 cm^2^ coupon occurred in less than 60 min under dry conditions [34]. The antiviral activities of ASO1 at 8 and 24 h and of ASO2 at 8 h post application were similar between the ISO21702 and ISO21702AD protocols. The antiviral activity of the active materials tested against influenza virus thus appears to be stronger when the total virus quantity within an inoculum is deposited dried on 1 cm^2^ than when the same quantity is spread over 16 cm^2^ or maintained in a liquid state. The antiviral activities of AS01 and AS02 were found to be stronger using the ISO21702 method against feline calicivirus compared to those measured using the NFS90700AD experimental procedure. The observed differences between the two protocols could be due to several factors, including the loss of virus on the reference surface and the potentially stronger action of ions released in the liquid inoculum on non-enveloped viruses [48]. Interestingly, observed differences between ISO21702AD and NFS90700AD protocols at 24 h post application suggested an impact of the inoculum area (16 vs. 1 cm^2^) on the antiviral activity, with the viruses being in contact with a greater quantity of active elements. As observed for the enveloped virus, the NFS90700AD protocol enabled the strongest antiviral activity of the copper alloy in AS03 against the non-enveloped calicivirus, compared to the other two protocols. To our knowledge, our study is the first aimed at directly comparing different standardized or modified experimental procedures to evaluate the antiviral activity of active materials. Our data show that experimental test conditions can have a substantial impact of over 1 log_10_ on the antiviral activity of the active material for the same contact period. The observed impact of these protocols on the antiviral activities directed towards enveloped and non-enveloped viruses should now be confirmed on a large panel of active materials and using several other viral strains of interest. Projections of biological fluids from an infected individual onto surrounding surfaces are generally small in terms of volume, tending to dry quickly. Considering that the antiviral action of self-decontaminating surface material depends, in real-life conditions, on micro-nanomaterial/virus interactions occurring within a dry inoculum, the NFS90700AD protocol thus seems the most relevant experimental approach to evaluate antiviral activity of an active material.

Laboratory simulations have demonstrated that hand-to-hand and fomite-to-hand contact are plausible modes of transmission for enteric viruses provided that the viral particles remain viable on hands and fomites in a sufficient quantity to reach the digestive tracts via finger contamination [13,14]. While the rate of hand-to fomite or fomite-to-hand transfer seems limited for enveloped viruses, several studies have described transfer rates from contaminated surfaces to hands corresponding to 15% for rhinovirus, from 23 to 34% for poliovirus, 16% for rotavirus and 22 to 38% for bacteriophages [14,15,49]. HITS including, for example, door handles, handrails, elevator buttons or electric switches can be contaminated by contact with hands soiled through inefficient hand washing of infected fecal deposits, in the case of gastroenteritis. Some studies have described the transfer of such viruses from soiled finger-pads to various food products, thus demonstrating the important role of food handlers in virus spread to foods and environmental surfaces that likely explains the higher numbers of foodborne outbreaks of noroviral gastroenteritis [4,6]. In our study, we implemented an approach allowing the transfer of norovirus, today believed to account for 65% of nonbacterial gastroenteritis outbreaks in the United States and Canada [50], from a contaminated soiled gloved finger-pad containing 10^5^ TCID50 norovirus, a concentration plausibly present on the finger of a contaminated person. We transferred the virus efficiently to both the reference surface and the copper alloy active material representing the HITS. The use of the contaminated gloved finger-pad allowed the transfer of around 10^4^ infectious norovirus particles onto the surfaces, corresponding to a finger/surface transfer rate of 4.4%, slightly lower than the 13% reported by Bidawid et al. with calicivirus [4]. The desiccation effect over time on stainless steel supports observed in the mixture virus/BSA transferred by contact with the finger-pad seemed to be stronger than that associated with the drying of the deposited viral solution (Figure 4). However, over 10^2^ infectious norovirus particles remained present on the recipient reference surface at 60 min post transfer. The minimal dose of norovirus required to cause human infection could be between 10 and 100 infectious viral units [4,11]. In a worst-case scenario in which only 5% of virus is transferred from a contaminated surface to finger-pads, at least 350 to 26 infectious virus particles would be transferred to the finger-pads via contact up to 15 min after the initial surface was contamination by transfer in our study, which most likely would be sufficient to initiate infection in susceptible individuals via contact between the contaminated finger and the mouth. Thus, the transfer rate of norovirus obtained in a reproducible manner using this approach appears sufficient to be considered as an innovative experimental approach to evaluate, in a viral gastroenteritis spreading context, the antiviral activity of active HITS contaminated by finger contact, a scenario highly representative of real-life conditions. The AS03 material was responsible for a rapid destruction of norovirus corresponding to over 2 log_10_ loss in the first 5 min post transfer, reaching the lower limit of quantification at 15 min post application. Considering the hypothesis of a 5% transfer efficiency of the virus through finger contact with a contaminated surface, the residual amount of norovirus on the active surface at 5 min post transfer was insufficient to allow the spread of the disease by finger contact. Thus, the theoretical transfer of the same quantity of virus, such as remains on a stainless-steel surface at 15 min post transfer, from AS03 onto a finger is only possible from 0 to 5 min post transfer. By comparing the amount of infectious virus on both reference steel and active surfaces, this experimental approach enables the determination of the antiviral activity of HITS that have been contaminated by contact under conditions closely resembling real life situations. The transfer via this approach of a mixture of virus and bacteria in soiled inoculum onto different surfaces has allowed the evaluation of antiviral activity of active HITS contaminated by a complex artificial mixture likely mimicking infectious microbial-containing fecal deposits transferred from contact with a soiled finger of an infected individual. Interestingly, the presence of the bacteria *Enterococcus faecium* present in human feces and used as a model in our study seemed to participate in the protection of norovirus by reducing the desiccation effect observed early on stainless steel supports. The protective role of bacteria on viruses has previously been described by Lopez et al. [13]. One could hypothesize, therefore, that the amount of norovirus on a surface up to 15 min following transfer through finger contact is greater in the presence of bacteria. For the AS03 material, the presence of bacteria on the active material had a moderate impact on the antiviral activity at 5 and 15 min post transfer; however, the residual amount of virus was significantly greater at the same time points in the presence of bacteria. The protective role of bacteria over viruses within an inoculum needs further investigation using combinations of bacteria and viruses that best represent those found in clinical samples. The use of a complex mixture offers the possibility of measuring simultaneously both antiviral and antibacterial activity of an active material following one transfer event. 

The experimental procedure proposed in our study and others offer the possibility of proposing a standardizable approach allowing the reproducible contamination of HITS in conditions resembling real-life via the transfer of infectious pathogens by finger contact. In response to the strong rise in the development of innovative self-decontaminating surface materials and in their promotion as a complementary strategy for disinfection to reduce the spread of microorganisms, our experimental approach should help meet the needs of production to test new surfaces in accordance with stringent end-user needs and requirements. We are currently studying the transfer of a mixture comprising enveloped respiratory virus (SARS-CoV-2 and influenza virus)/mucus nasal/saliva in the presence or not of *Streptococcus pneumoniae* by finger-surface contact using the proposed approach and we are developing an innovative technology combining autofluorescent viruses and living human skin explants for the evaluation of antiviral activity of active materials.

## Figures and Tables

**Figure 1 materials-16-02889-f001:**
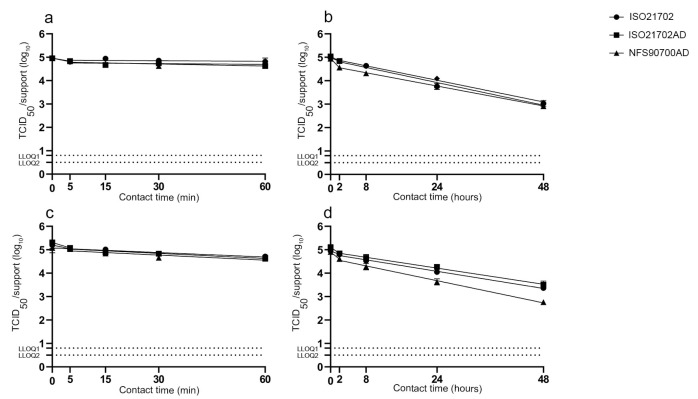
Survival of enveloped H3N2 virus (**a**,**b**) and non-enveloped feline calicivirus (**c**,**d**) on stainless steel reference surfaces according to ISO21702, ISO21702AD and NFS90700AD protocols. LLOQ: lower limit of quantification (LLOQ2 = 0.5 log_10_ TCID_50_ for 21702/21702AD methods and LLOQ1 = 0.8 log_10_ TCID_50_ for NFS90700AD method). Value displayed as average +/−SD.

**Figure 2 materials-16-02889-f002:**
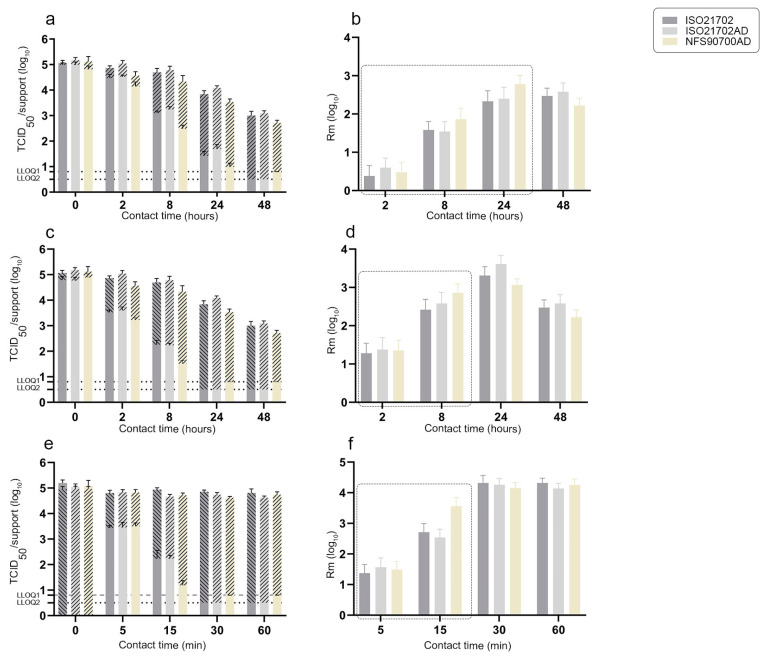
Amount of infectious H3N2 influenza virus at 0, 2, 8, 24 and 48 h post application (**a**,**c**) or at 0, 5, 15, 30 and 60 min post application (**e**) on both active surfaces (plain box) and stainless steel supports (hatched box) according to ISO21702, ISO21702AD and NFS90700AD protocols (mean ± SD, 9 supports/condition). Average antiviral activity Rm (log_10_ TCID_50_/cm^2^) including its 95% confidence interval (Kr, 3 independent experiments) according to ISO21702, ISO21702AS and NFS90700AD protocols. (**a**,**b**): AS01; (**c**,**d**): AS02; (**e**,**f**): AS03. LLOQ: lower limit of quantification (LLOQ2 = 0.5 log_10_ TCID_50_ for the ISO21702/ISO21702AD protocols and LLOQ1 = 0.8 log_10_ TCID_50_ for the NFS90700AD protocol).

**Figure 3 materials-16-02889-f003:**
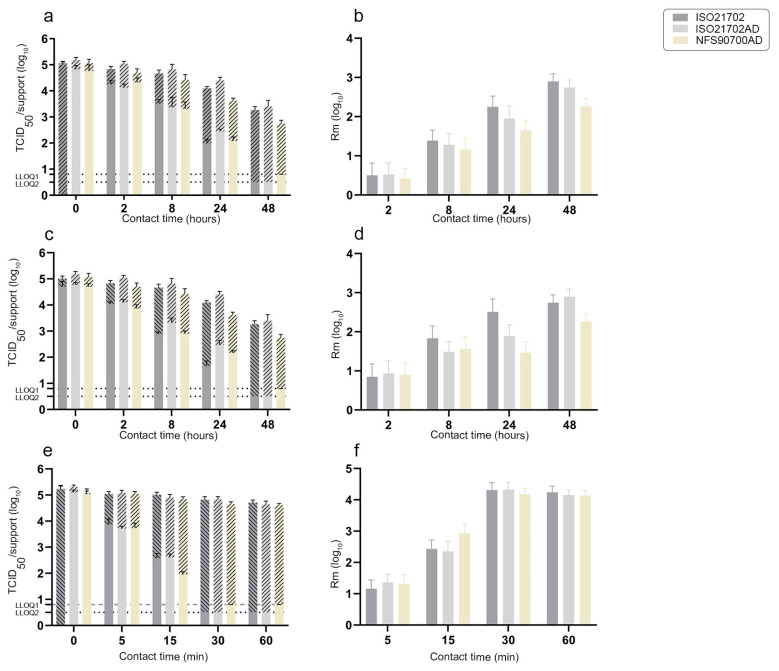
Amount of infectious feline calicivirus at 0, 2, 8, 24 and 48 h post application (**a**,**c**) or at 0, 5, 15, 30 and 60 min post application (**e**) on both active surfaces (plain box) and stainless steel supports (hatched box) using ISO21702, ISO21702AS and NFS90700AD protocols (mean ± SD, 9 supports/condition). Average antiviral activities Rm (log_10_ TCID_50_/cm^2^), including 95% confidence interval (Kr, 3 independent experiments) using ISO21702, ISO21702AS and NFS90700AD protocols (**b**,**d**,**f**). (**a**,**b**): AS01; (**c**,**d**): AS02; (**e**,**f**): AS03. LLOQ: lower limit of quantification (LLOQ2 = 0.5 log_10_ TCID_50_ for the ISO21702/ISO21702AD protocols and LLOQ1 = 0.8 log_10_ TCID_50_ for the NFS90700AD protocol).

**Figure 4 materials-16-02889-f004:**
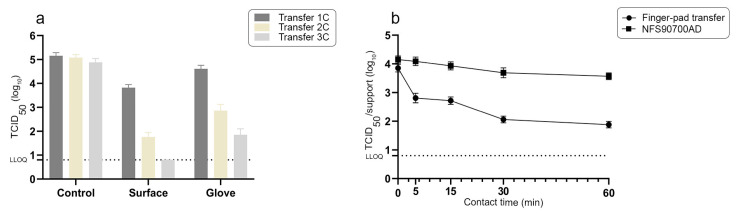
(**a**): Amount of infectious murine norovirus at T0 transferred from artificially contaminated gloved fingertip to stainless steel supports (surface); residual quantities of virus on gloved fingertip after the viral transfer (glove) and from direct washing of fingertips prior to transferring to the surfaces (control). Transfers 1C, 2C and 3C correspond to data obtained after the 1st contact, the 2nd contact and the 3rd contact, respectively. Mean values ± SD corresponding to 10 independent viral transfer experiments for each condition (**b**): Survival of murine norovirus on stainless steel surfaces over time after inoculation by finger-pad viral transfer (first contact) or by a viral solution deposit (NFS90700AD procedure). LLOQ: lower limit of quantification (LLOQ = 0.8 log_10_ TCID_50_). Mean values ± SD corresponding to 10 inoculated supports for each time point.

**Figure 5 materials-16-02889-f005:**
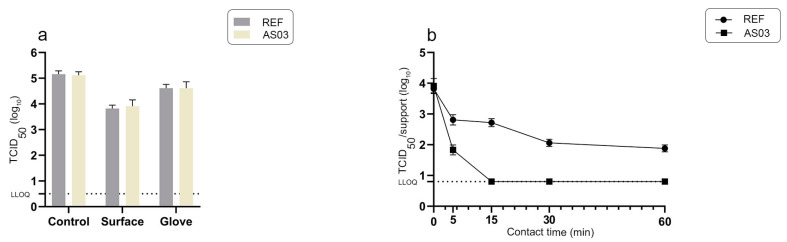
(**a**): Amount of infectious murine norovirus at T0 transferred from the artificially contaminated gloved fingertip to stainless steel supports (REF) and AS03 active surface (surface); residual quantities of virus on gloved fingertip after the viral transfer (glove) and from direct washing of fingertips prior to transferring to the surfaces (control). Mean values ± SD corresponding to 10 independent viral transfers for each surface. (**b**): Survival of murine norovirus on stainless steel supports (REF) and AS03 active surface after finger-pad viral transfer. LLOQ: lower limit of quantification (LLOQ = 0.8 log_10_ TCID_50_). Mean values ± SD corresponding to 10 inoculated supports for each time point.

**Figure 6 materials-16-02889-f006:**
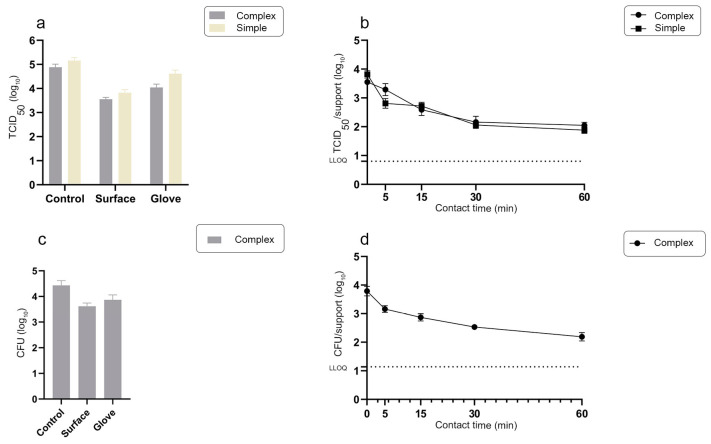
(**a**): Amount of infectious murine norovirus on stainless steel supports (surface) at T0, following transfer from a gloved fingertip artificially contaminated with the virus alone (simple) or with the complex mixture (complex); residual quantities of virus on gloved fingertip after the viral transfer (glove) and from direct washing of fingertips prior to transferring to the surface (control). Mean values ± SD corresponding to 10 independent viral transfers. (**b**): Survival of murine norovirus on stainless steel supports after viral transfer (simple) or mixture transfer (complex). LLOQ: lower limit of quantification (=0.8 log_10_ TCID_50_). Mean values ± SD corresponding to 10 inoculated supports for each time point. (**c**): Amount of *Enterococcus faecium* bacteria at T0 on stainless steel supports (surface) transferred from gloved fingertip artificially contaminated with complex mixture; residual quantities of bacteria on gloved fingertip after the transfer (glove) and from the direct washing of fingertips prior to transferring to the surface (control). Mean values ± SD corresponding to 10 independent experimental transfers. (**d**): Survival of *Enterococcus faecium* bacteria on stainless steel supports after mixture transfer (complex). LLOQ: lower limit of quantification (=1.1 log_10_ CFU corresponding to 14 colonies). Mean values ± SD corresponding to 10 inoculated supports for each time point.

**Figure 7 materials-16-02889-f007:**
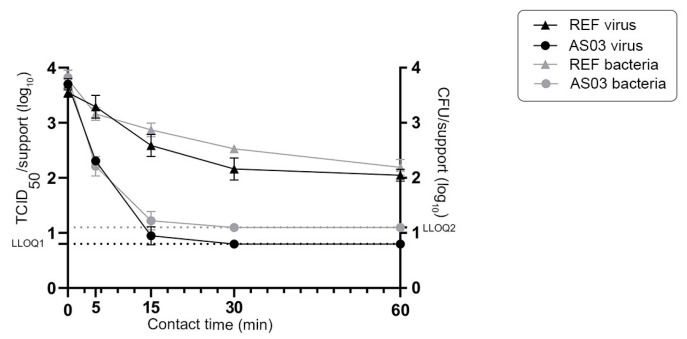
Survival of both murine norovirus and *Enterococcus faecium* on stainless steel and AS03 supports after mixture transfer. LLOQ: lower limit of quantification (LLOQ1 = 0.8 log_10_ TCID_50_ and LLOQ2 = 1.1 log_10_ CFU). Mean values ± SD corresponding to 10 inoculated supports for each time point.

## Data Availability

Not applicable.

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
