# Peer review of "Antiviral Activity of Active Materials: Standard and Finger-Pad-Based Innovative Experimental Approaches"

_materials, 2023, doi:10.3390/ma16072889_

Round 1

Reviewer 1 Report

The paper reports on the comparative study of several materials possessing antiviral properties. These materials are intended for use in elaboration of high-touch surfaces preventing fomite-to-hand transmission of viral infections. The subject of the paper falls within the scope of Materials journal. The results obtained are new and worth of being published. The authors demonstrated possible directions of antiviral materials development and, more importantly, evaluated available techniques of testing these materials. Several protocols of antiviral activity assessment were compared in terms of possible practical applications, the differences between these protocols were revealed and discussed.

I have the following comments:

1. The information on the active materials provided in the section "Active and reference materials" is too scarce. Please provide some additional details on the chemical and phase composition of these materials, not only the information on their manufacturers.

2. Please justify the choice of the active materials tested.

3. Please replace "student" with "Student" (line 213).

Reviewer 2 Report

The paper by Szpiro et al. presents in a very meticulous manner the role of high-touch surfaces in transmission of viral/bacterial infections. The  work is in my opinion very good in methodological terms. It uses different protocols, with modified conditions aiming to mimic the environmental conditions in the real life. It clearly shows usefulness of one protocol over the other two. The strongest point is the use of a complex mixture (Norovirus and Enterococcus feacalis) to show that bacteria can support the persistence of viruses on the specific surfaces. However, some parts of the paper rise questions:

1) introduction - line 94 - the authors point that silver inhibits several viruses. However, the references are not proper (26 and 27), since the mentioned papers do not decribe silver activity aginst the listed viruses.

2) Figure 2 and 3 are difficult  to read - please make them larger or just more clear.

3) What was the purpose of using Rm parameter? Can you describe it?

4) Discussion - 540 - the statement on the concentration of Ag ions is speculation. We can actually measure the amount of released ions in wash-outs (ASA or ICP-MS).
